

# A GLUE-based assessment of WaTEM/SEDEM for simulating soil erosion, transport, and deposition in soil conservation optimised agricultural watersheds

Kay D. Seufferheld[1], Pedro V. G. Batista[1], Hadi Shokati[2], Thomas Scholten[2], Peter Fiener[1]

[1]Institute of Geography, Water and Soil Resources Research, University of Augsburg, Augsburg, 86159, Germany.
[2]Department of Geosciences, Soil Science and Geomorphology, University of Tübingen, 72074, Tübingen, Germany.

*Correspondence to:* Peter Fiener (peter.fiener@geo.uni-augsburg.de, Alter Postweg 118, 56159 Augsburg, Germany)

**Abstract.** Soil erosion models are essential tools for soil conservation planning. Although these models are generally well-tested against plot and field data for in-field soil management, challenges arise when scaling up to the landscape level, where sediment trapping along landscape features becomes increasingly critical. At this scale, a separate analysis of model performance in representing erosion, sediment transport, and deposition processes is both challenging and often lacking. In this study, we assessed the capacity of the spatially distributed erosion and sediment transport model WaTEM/SEDEM to simulate sediment yields in six micro-scale watersheds ranging from 0.8 to 7.8 ha, monitored over eight years from 1994 to 2001. The watersheds were comprised of two groups: four field-dominated watersheds characterised by arable land with minimal landscape structures, and two structure-dominated watersheds featuring a combination of arable land and linear landscape structures (mainly grassed waterways along thalwegs) that minimise sediment connectivity. This setup enabled a separate analysis of model performance for both watershed groups. A Generalised Likelihood Uncertainty Estimation (GLUE) framework was employed to account for measurement and model uncertainties across multiple spatiotemporal scales. Our results show that while WaTEM/SEDEM generally captured the magnitude of the very low measured sediment yields in the monitored watersheds, the model did not meet our pre-defined limits of acceptability when operating on annual timesteps. However, the WaTEM/SEDEM´s performance improved substantially when model realisations were aggregated across the eight-year monitoring period and over the two watershed groups, with mean absolute errors of 0.11 t ha$^{-1}$ yr$^{-1}$ for field-dominated and 0.18 t ha$^{-1}$ yr$^{-1}$ for structure-dominated watersheds. Our findings demonstrate that the model can represent the influence of soil conservation measures on reducing soil erosion and sediment delivery but performs better for long-term conservation planning at larger scales than for precise annual predictions in individual micro-scale watersheds with specific conservation practices.



**1. Introduction**
Soil erosion by water is a major threat to global soil health and associated ecosystem functions and services,
endangering agricultural sustainability and food security (Rickson et al., 2015; Montanarella et al., 2016; Quinton
and Fiener, 2024). Although the problem of accelerated soil erosion has been known for a long time and a wide
variety of soil conservation practices have been tested and implemented locally for many decades, adoption
remains limited due to economic constraints, lack of technical knowledge, and insufficient policy support
(Quinton and Fiener, 2024; Aghabeygi et al., 2024). This is particularly problematic in regions where the
intensification of agriculture, exemplified by the historical increase in the size and weight of agricultural
machinery that has led to increased soil compaction levels (Brus and Van Den Akker, 2018; Keller et al., 2019),
and the increase in frequency and intensity of extreme precipitation events due to climate change (Auerswald
and Fiener, 2024; Hosseinzadehtalaei et al., 2020; Myhre et al., 2019) is likely exacerbating the erosion risk.
Effective soil conservation relies on two complementary strategies: (i) In-field control measures that increase soil
surface cover by vegetation and hence prevent soil detachment by raindrop impact and sheet flow. Such
measures include optimised crop rotations, using cover crops, and soil residue management (Andersson and
D'souza, 2014). (ii) Off-site sediment transport control structures along the runoff pathway that increase
infiltration and foster sediment trapping and minimise sediment connectivity. Typical structures are vegetative
filter strips (Gumiere et al., 2011), grassed waterways (Fiener and Auerswald, 2003), retention ponds (Fiener et
al., 2005), or a generally optimised layout of fields along slopes (Van Oost et al., 2000).
Soil erosion models are potentially valuable tools for identifying high erosion risk areas and evaluating
intervention needs, enabling stakeholders to effectively implement soil conservation strategies. Diverse models
have been developed and applied for this purpose, ranging from empirical and conceptual to process-oriented
model types (e.g. Eekhout et al., 2018; Smith et al., 2018; Nearing, 2013; Dymond et al., 2010; Hessel and Tenge,
2008). The most widely used model for soil conservation planning is the Universal Soil Loss Equation (USLE)
(Wischmeier and Smith, 1978) and its revisions and regional adaptations, like the revised USLE (RUSLE) (Renard,
1997) and the German ABAG (Allgemeine Bodenabtragsgleichung, German for Universal Soil Loss Equation; Din-
Normenausschuss, 2022; Schwertmann et al., 1987).
While these USLE-type models have been adapted to calculate spatially distributed erosion rates, they are limited
to calculating potential soil loss without considering sediment transport processes and downslope deposition. To
overcome this limitation, the Water and Tillage Erosion Model and the Sediment Delivery Model
(WaTEM/SEDEM) (Van Oost et al., 2000; Van Rompaey et al., 2001; Verstraeten et al., 2002) was developed.
WaTEM/SEDEM combines the RUSLE (Renard, 1997) with spatially distributed sediment transport and deposition
modelling. The performance of the model has been tested using sediment trapping in reservoirs (e.g. Hlavčová
et al., 2018), sediment delivery in small rivers of mesoscale catchments (e.g. Batista et al., 2022; Rehm and Fiener,
2024), or long-term erosion and deposition patterns derived from radionuclides (e.g. Van Oost et al., 2000; Wilken
et al., 2020). However, to the best of our knowledge, the suitability of WaTEM/SEDEM for representing soil
erosion, transport, and deposition processes within soil conservation settings combined with measures to reduce
sediment connectivity, which can minimize sediment redistribution, has not been thoroughly tested.



Testing the ability of spatially distributed erosion models to simulate the combined effects of in-field soil
conservation and landscape features trapping sediments is inherently challenging. Observational data for model
calibration and validation are typically restricted to measurements of sediment yields at the outlet of a system
(Batista et al., 2019), which typically consist of small erosion plots, meso-scale watersheds, or large-scale
catchments. Such outlet-based measurements do not allow for testing a model's representation of internal
erosion and deposition patterns, as they provide little information on the spatial distribution of sediment sources
and sinks within the landscape. This exacerbates the equifinality problem (Beven, 2006), and models may achieve
accurate outputs while incorrectly representing the spatial patterns of erosion and deposition processes within
watersheds.
Micro-scale watersheds (1-10 ha) are ideal for evaluating soil conservation measures typically implemented from
the field to the landscape level (Choudhury et al., 2022; Fiener and Auerswald, 2018). This is because soil erosion
and sediment connectivity processes that are distinguishable at the micro-scale watershed are not represented
in small plots or get diluted in large-catchment sediment yield observations. Moreover, important input data for
erosion modelling, e.g. rainfall, soil management, and land cover, can be monitored and measured with higher
detail at the micro-scale, compared to larger areas (Fiener et al., 2019a). Nevertheless, there is limited research
on modelling the combined effects of in-field soil conservation and landscape structures on soil redistribution
and sediment delivery at this scale.
Notwithstanding the spatial extent of (long-term) soil erosion monitoring, measurement uncertainties arise from
instrumental precision and temporal instrument malfunctioning, data handling and processing. The uncertainties
in observational data have important implications for erosion modelling, as models cannot be expected to be
better than the observational data (Beven and Lane, 2022; Beven, 2019).
The Generalized Likelihood Uncertainty Estimation (GLUE) framework (Beven and Binley, 1992) allows for testing
environmental models while accounting for the uncertainty in both models and the observational data. In light
of inherent measurement uncertainties, GLUE acknowledges that it is not possible to identify a single parameter
set as "correct". Rather, all parameter combinations that produce results within the observational uncertainty
cannot be rejected. Within the GLUE framework, limits of acceptability are defined to identify which model runs
fall within the uncertainty bounds of the measurements (Beven and Lane, 2022). These behavioural models are
retained, while non-behavioural models are rejected. This limits-of-acceptability GLUE approach thus provides a
systematic methodology to evaluate model performance with uncertain testing data.
In this study we employ this limit-of-acceptability approach based on the GLUE framework, focusing on three
main objectives: (i) testing WaTEM/SEDEM's capability to simulate sediment yields in micro-scale watersheds
either characterised by in-field soil conservation or by in-field soil conservation plus linear landscape features
designed to trap sediments, (ii) analysing the behaviour of model parameters that control erosion and sediment
transport processes, and (iii) assessing the model's performance across different spatiotemporal resolutions
through data aggregation. We accomplish these objectives using a comprehensive dataset from a long-term,
farm-scale monitoring in Southern Germany, which provides continuous precipitation, surface runoff and



sediment flux data from six micro-scale watersheds under optimised soil conservation (Auerswald et al., 2001;
Auerswald and Fiener, 2019).
**2. Material and methods**
**2.1 Test site**
The test site is part of an experimental farm located in Scheyern, southern Germany (48°29'45.1"N, 11°26'23.6"E;
about 470 m above sea level). It is part of Bavaria's tertiary hill region, an important and productive agricultural
landscape in Central Europe. The rolling topography is characterised by predominantly east-facing slopes ranging
from 0.4° to 11.5° (Wilken et al., 2019a). Climate conditions include a mean annual temperature of 8.4 °C and
mean annual precipitation of 834 mm (1994-2001), with the highest precipitation occurring between May and
July (Fiener et al., 2019a). Management practices at the farm follow a comprehensive soil conservation
philosophy based on two main principles: (i) keeping arable soils covered as long as possible and (ii) reducing
hydrological and sedimentological connectivity as far as possible (Fiener et al., 2019a). Within the watersheds,
soils consist predominantly of loamy or silty loamy Cambisols (World Reference Base for Soil Resources (WRB),
Schad et al., 2022).
The research area comprises six micro-scale watersheds (W01-W06) with a total area of 24 ha and four
agricultural fields (F15-F18, Fig. 1). The six watersheds exhibit different landscape connectivity characteristics:
W01-W04 (0.8 to 4.2 ha) are classified in this study as field-dominated systems due to their structure, with most
of their area covered by agricultural fields and minimal landscape structures along sediment flux pathways. In
contrast, W05 and W06 are classified as structure-dominated systems due to their configuration, featuring more
complex landscapes. The Watershed W06 (5.7 ha) constitutes the upper part of the larger watershed W05 (7.8
ha) (Fiener et al., 2019a).
Three key conservation measures were implemented to minimise hydrological and sedimentological connectivity:
(i) optimised field layout with fields arranged parallel to contour lines, (ii) retention ponds at field borders, and
(iii) a grassed waterway along the main thalweg of W05 and W06. The retention ponds were located at the outlets
of watersheds W01, W02, W05, and W06 (Fig. 1). Sediment trapping efficiency measurements were conducted
for these ponds, revealing an average of 70 ± 14 % (Fiener et al., 2005). Additionally, continuous monitoring
systems were installed at the outlet of each micro-scale watershed to measure runoff and sediment delivery. The
distinction between field-dominated and structure-dominated watersheds will be used consistently throughout
this study.
All fields within the watersheds were managed using no-till practices with a crop rotation of winter wheat, maize,
winter wheat, potatoes, whereas the rotation was shifted between the fields (F15-F18, Fig. 1). After winter wheat,
mustard was sown as a cover crop. In the case of potatoes, the mustard was sown into the potato dams built in
autumn, while direct seeding into the down-frozen mustard was performed in the following year.





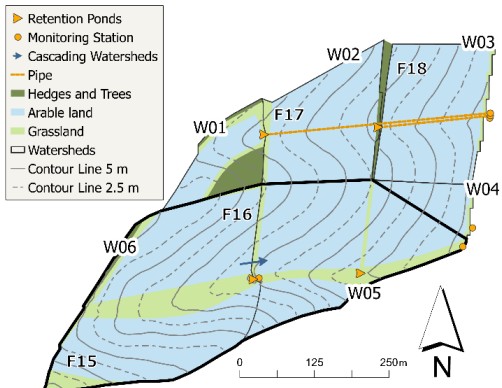


**Figure 1: Land use and topography of the experimental farm in Scheyern, Bavaria, with flow direction from west to east.**
**Note: watershed W05 (thick line) includes the upslope watershed W06.**
**2.2 Data**
The study utilised a unique erosion monitoring dataset acquired between 1994 and 2001. This comprehensive
dataset, as well as metadata, are provided by Fiener et al. (2019a). All spatial data were resampled to a consistent
5 m by 5 m grid resolution, matching the digital elevation model (DEM) provided in the dataset (Wilken et al.,
2019a). The temporally dynamic input data included daily soil cover measurements and high-resolution
precipitation data recorded at 1-minute intervals from up to 11 monitoring sites (Wilken et al., 2019b). Additional
details regarding these input parameters are provided in section 2.4 below.
For model testing, we used continuous sediment delivery data from the six micro-scale watersheds between 1994
and 2001. Runoff and suspended-sediment loads were monitored with a measuring system based on a
Coshocton-type wheel sampler (precision ± 10%; Carter and Parsons, 1967; Fiener and Auerswald, 2003). The
device continuously diverted an aliquot of approximately 0.5 % from the total flow that left the watersheds
through underground-tile outlets with a diameter of 15.6 cm and 29 cm (Fig. 1). At lower rates (< 0.5 L s$^{-1}$) the
system slightly over-estimated runoff, but these small events contributed negligibly to the cumulative water and
sediment budgets. Under sampling during very high flows was avoided by (i) employing large wheels (Ø 61 cm)
and (ii) the flow-dampening effect of the retention ponds situated immediately upstream of each outlet (Fiener
and Auerswald, 2003).
**2.3 Soil erosion modelling**
The WaTEM/ SEDEM version used in this study consists of two main components: (i) WaTEM, which implements
a spatially distributed German adaption of the USLE , and (ii) SEDEM, which incorporates a transport capacity (*TC*)
equation (Eq. 3) and a routing algorithm for sediment re-distribution based on a DEM (Verstraeten et al., 2002;
Van Rompaey et al., 2001; Van Oost et al., 2000). To implement WaTEM/ SEDEM within the GLUE-framework, the
original Delphi code-based model was translated to Python 3.12 and was run in PyCharm 2024.1 (Community
Edition), which substantially improved computational speed through parallel processing and allowed for easier



data handling. Although the Python implementation includes tillage erosion calculations, this component was not
utilised in the present study.
The model was applied for the period from April to October of each year from 1994 to 2001, excluding periods
potentially affected by snowmelt erosion and prolonged surface runoff from return flow (Fiener et al., 2019a).
While these months contributed 10.7 % of the total measured sediment delivery (Fiener et al., 2019b), our
analysis focused on the dominant water erosion period during heavy rainfall months. Each micro-scale watershed
was separately modelled.
**2.4 Potential Erosion**
In contrast to the original WaTEM/ SEDEM (Verstraeten et al., 2002; Van Rompaey et al., 2001; Van Oost et al.,
2000), in which the USLE factors are derived according to the RUSLE approach (Renard, 1997), we calculated the
USLE factors as calculated according to their German adaptation (Eq. 1) (Schwertmann et al., 1987; Din-
Normenausschuss, 2022):
$A = R * K * LS * C * P,$                                                   (1)
Where $A$ is the potential erosion (t ha$^{-1}$ yr$^{-1}$), $R$ the rainfall erosivity factor in (N h$^{-1}$ yr$^{-1}$), $K$ the soil erodibility
factor (t ha$^{-1}$ h N$^{-1}$), $LS$ the slope length and steepness factor (dimensionless), $C$ the cover management factor
(dimensionless), and $P$ the agricultural practices factor (dimensionless).
The high-resolution rainfall data from eleven (1994–1997) and two (1998–2001) precipitation monitoring stations
located in the research area were used to calculate the rain erosivity factor (R-factor) (Wilken et al., 2019b).
According to the German adaptation of the USLE, rainfall events were considered erosive if they met at least one
of two criteria: (i) total rainfall amount ≥ 10 mm or (ii) maximum 30-minute intensity ≥ 10 mm h$^{-1}$. Individual
events were separated by at least 6 hours without rainfall (Schwertmann et al., 1987; Din-Normenausschuss,
2022). The calculated rainfall erosivities per monitoring station were interpolated to 5 m by 5 m resolution maps
using inverse distance weighting, and the spatially distributed values ranged between 65.90 and 155.10 N h$^{-1}$ yr$^{-1}$
$^{1}$ across the eight-year study period.
Soil erodibility (K factor) values were computed following Auerswald et al. (2014) and already provided in the
monitoring data set (Auerswald et al., 2019a). The values, originating from a 50 by 50 m sampling grid, were
spatially interpolated using ordinary kriging to generate a continuous surface with a 5 m by 5 m resolution grid.
The resulting K factor values across the study area ranged from 1.8 to 4.6 t ha$^{-1}$ h N$^{-1}$.
The slope length and slope steepness factor (LS factor) was calculated based on the DEM using the approach by
Desmet and Govers (1996). When calculating the LS factor for W01, the shrubbed area (Fig. 1) was excluded due
to its negligible runoff contribution. Additionally, we calculated the LS factors for W02 and W03 separately from
their upslope catchments (i.e. W01 and W02), since their runoff was directed via underground pipes to the
monitoring stations (see Fig. 1).
The annual crop factor (C factor) was calculated by combining seasonal rainfall erosivity with temporal changes
in soil coverage (Schwertmann et al., 1987). The soil loss ratio (*SLR*) quantifies the protective effect of soil



coverage by comparing potential soil loss under a given vegetation condition to that under standardised fallow
conditions (Schwertmann et al., 1987; Wischmeier and Smith, 1978). While the *SLR* traditionally considers five
crop growth stages, from bare soil (0% cover) to full canopy coverage (75-100% cover), we also considered crop
residue cover.
From 1994 to April 1997, direct bi-weekly measurements during growing seasons and monthly measurements
during autumn and spring were conducted, with additional observations before and after soil management
operations. These field measurements included both crop and residue cover. From these field measurements,
standardised daily crop development and residue cover were established and used for the subsequent period
from April 1997 onwards (Auerswald et al., 2019b; Fiener et al., 2019a).
Total soil cover was calculated with residues protecting portions of the otherwise exposed soil according to:
$$Co_{tot} = Co_{crop} + (100 - Co_{crop}) * \frac{Co_{res}}{100},$$  (2)
With $Co_{tot}$ is the total soil cover (%), $Co_{crop}$ the cover of the growing crop on the respective field (%), and $Co_{res}$
the measured soil cover of the residues (%).
Figure 2 illustrates the total soil cover on the respective fields with monthly rainfall erosivity. Determining field-
specific *SLR* values involved categorising soil cover into the five growth stages and assigning corresponding *SLR*
values. As no-till was applied at the research farm, lower *SLR* values were assigned than in conventional systems
due to increased soil surface protection. These *SLR* values were obtained from Schwertmann et al. (1987) and
adapted based our expert knowledge regarding the soil conservation practices in the Scheyern experimental farm
(Fiener and Auerswald, 2007; Fiener et al., 2019a).





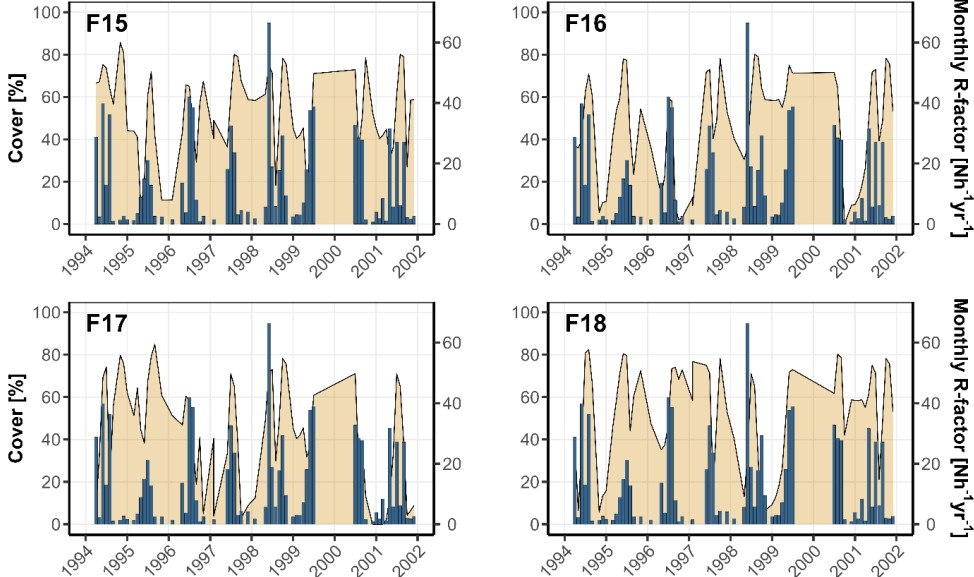


**Figure 2: Each field's total soil cover (Residues and crops). Blue bar plots (monthly sum) show monthly R-factors.**

The support practices factor (P factor) was not specifically parametrised for contour-seeding because of field
heterogeneity, i.e. not all parts of a single field were contour-seeded, and/or the absence of specific P factor
values for structures such as the potato dams. However, we accounted for the uncertainty stemming from this
lack of parameter representation as part of the model conditioning process (see section 2.4 below).
**2.5 Sediment Transport and Deposition**
The Transport Capacity (*TC*) quantifies the maximum amount of sediment transported through a grid cell without
deposition. When the incoming sediment load into a raster cell exceeds TC, the excess material is deposited within
the cell, whilst the remaining portion continues its downstream movement. *TC* was calculated with the approach
proposed by Van Rompaey et al. (2001):
$$TC = k_{TC} * R * K * (LS - S_{IR}),\qquad(3)$$
*with:*
$$S_{IR} = 4.12 * S_m^{0.8}\qquad(4)$$
where $TC$ is the transport capacity (t ha$^{-1}$), $k_{TC}$ the transport capacity coefficient (m) described below, $R$ the
rainfall erosivity factor in (N h$^{-1}$ yr$^{-1}$), $K$ is the soil erodibility factor (t ha$^{-1}$ h N$^{-1}$), $LS$ the slope length and steepness
factor (dimensionless), $S_{IR}$ the interrill slope gradient factor (dimensionless) and $S_m$ the slope (m m$^{-1}$).
The transport capacity coefficient ($k_{TC}$) represents the theoretical upslope distance required for sediment
generation to reach maximum *TC* at a given raster cell under the assumption of uniform slope and erosion
conditions (Van Rompaey et al., 2001). The transport capacity coefficient depends on surface roughness and



therefore differs according to land use and management. In our model parameterisation, we distinguish between
higher values for arable ($k_{TC/A}$) land and lower values for grassland ($k_{TC/G}$; along field borders and in grassed
waterways).
WaTEM/SEDEM's hillslope sediment transport module employs a multiple flow routing algorithm, which
distributes sediment from individual cells to their downslope neighbours based on Quinn et al. (1991). The
algorithm calculates local slopes to eight neighbouring cells and applies specific weighting factors: 0.50 for
orthogonal neighbours and 0.35 for diagonal neighbours. The sediment flux is distributed proportionally to the
weighted slope values of all cells at equal or lower elevations.
In this study, we implemented the Parcel Connectivity ($p_{con}$) parameter specifically at field boundaries. $p_{con}$
reduces the contributing upstream area by a value [%] at these transitions (Notebaert et al., 2006). This reduction
has a dual effect: (i) it directly lowers the slope length part of the LS factor, thereby decreasing the potential
erosion for subsequent downstream cells, and (ii) it affects the TC, which is calculated using the LS factor (Eq. 3).
Unlike the original WaTEM/SEDEM version (Notebaert et al., 2006), we implemented $p_{con}$ within the multiple flow
routing algorithm loop calculating the contributing upstream area, ensuring its effects propagate downstream
through the flow network. Consequently, the reduction in sediment transport influences the downstream cells
and extends to subsequent agricultural fields and vegetated areas. Moreover, we introduced a border deposition
($b_{dep}$) parameter, which represents a forced deposition mechanism activated when agricultural field cells
contribute sediment to adjacent vegetated areas. Under these conditions, a defined percentage of the
transported sediment is deposited directly at the field border within the field.
Retention ponds were implemented within the 5 m by 5 m land use raster map. The locations of the four retention
ponds at the outlets of the micro-scale watersheds were mapped, with assigned trapping efficiencies of 54 %, 82
%, 59 %, and 85 % for watersheds W01, W02, W05, and W06, respectively, as measured in Fiener et al. (2005).
The standard deviation across all watersheds (± 13.7 %) was applied to account for measurement error in the
trapping efficiency values.
**2.6 Generalised Likelihood Uncertainty Estimation (GLUE) framework**
We employed the GLUE methodology (Beven and Binley, 1992) to represent model and measurement
uncertainties and to identify and analyse behavioural parameter sets. The GLUE approach recognises that
multiple parameter sets may provide equally acceptable simulations of a system within the limitations of a given
model structure and observational errors (Beven, 2006).
We established limits of acceptability for the simulated sediment yields by considering multiple sources of
uncertainty in the event-based measurements of runoff and sediment concentrations used for calculating annual
and median annual sediment yields. These included Coshocton wheel measurement errors (± 10%, Fiener and
Auerswald, 2003), runoff collector barrel sampling errors (estimated ± 10 %), and retention pond uncertainties (±
14 %). For events with data collection issues (flagged in the data set), we assigned an additional ± 50 % error
margin. However, for events flagged as "barrels overflown", we introduced only an upper error boundary since
the measurement taken from the barrel represents a minimum possible sediment yield during a rainfall event.



Finally, we propagated the measurement errors using a Monte Carlo simulation with 1,000 realisations and
sampling from normal distributions that represented the range of potential errors. The 2.5[th] and 97.5[th] percentiles
of the resulting aggregated (annual and median annual) sediment yields were used as the limits of acceptability
for simulated values. These uncertainty bounds served as criterion for behavioural model realisations. Hence,
only simulations producing outputs within these error margins were classified as behavioural and retained for
subsequent analysis.
**2.7 Model evaluation**
The model results were evaluated using R-Studio (R 4.4.2; R-Studio 2024.12.1 Build 563) in two phases to account
for the different sediment transport processes in field-dominated and structure-dominated watersheds.
**Phase 1 - Field-dominated watersheds:**
We performed a Monte Carlo simulation with 25,000 realisations for the field-dominated watersheds, sampling
parameters from uniform distributions across *a priori* selected ranges (Tab. 1). To consider the inherent potential
errors in USLE calculations, including uncertainties associated with the parameterisation of the P factor, we
modified the potential erosion in individual raster cells through an error surface ($e_{sur}$) before routing the
sediment. This error surface was sampled from a uniform distribution for each realisation, modifying the USLE-
calculated potential erosion (Eq. 1) within a range of 0 to ± 0.5:
$A_{new,i} = A_i + A_i * e_{sur},$ (5)
Where $A_i$ is the potential soil erosion (t ha$^{-1}$ yr$^{-1}$) calculated by the USLE (Eq. 1) at raster cell $i$, $A_{new,i}$ is the
potential soil erosion (t ha$^{-1}$ yr$^{-1}$) with incorporated uncertainty at raster cell $i$, and $e_{sur}$ the error surface
(dimensionless).
To ensure that $k_{TC/G}$ is consistently lower than $k_{TC/A}$, both were sampled with a constrained relationship, where
$k_{TC/A}$ values were required to be at least 1.5 but no more than 5 times higher than $k_{TC/G}$ values. Model runs were
classified as behavioural if the simulated sediment yield values fell within the established limits of acceptability
for the observed data. For these behavioural simulations, we calculated likelihoods by rescaling the mean
absolute error (*MAE*) (Brazier et al., 2000):
$L_i = \frac{1}{MAE_i} / \sum \frac{1}{MAE_i},$ (6)
*with:*
$MAE_i = |Sim_i - Obs_i|,$ (7)
where $L_i$ is the likelihood of one realisation $i$ (dimensionless), $MAE_i$ is the mean absolute error of realisation $i$ (t
ha$^{-1}$ yr$^{-1}$), $Sim_i$ is the simulated values for behavioural runs of realisation $i$ (t ha$^{-1}$ yr$^{-1}$), and $Obs_i$ is the observed
sediment value for realisation $i$ (t ha$^{-1}$ yr$^{-1}$).





**Table 1: Parameter ranges used for MC simulation in the WaTEM/SEDEM model. These ranges were selected based on the**
**literature on previous model applications. $k_{TC/A}$ and $k_{TC/G}$ are the transport capacity coefficients for arable land and**
**grassland, $p_{con}$ is the parcel connectivity, $e_{sur}$ is the error surface and $b_{dep}$ the border deposition.**

| Range | $k_{TC/A}$ [m] | $k_{TC/G}$ [m] | $p_{con}$ [%] | $e_{sur}$ | $b_{dep}$ [%] |
|---|---|---|---|---|---|
| low | 1 | 1 | 50 | -0.5 | 0 |
| high | 300 | 100 | 90 | 0.5 | 20 |

**Phase 2 - Structure-dominated watersheds:**
For the structure-dominated watersheds, we used the likelihoods associated with behavioural parameter values
conditioned in Phase 1 to represent in-field processes ($k_{TC/A}$ and $e_{sur}$) in order to generate another model 25,000
realisations. In this second phase, the model conditioning was focused on the parameters controlling sediment
redistribution through landscape structures ($k_{TC/G}$, $b_{dep}$ and $p_{con}$). The same limits of acceptability approach as in
phase one was applied to identify behavioural simulations. We calculated new likelihood values for these
simulations to analyse their performance in representing structural erosion control measures.
**2.8 Spatiotemporal model evaluation**
Model outputs were analysed at multiple spatiotemporal scales through sequential aggregation steps: First, we
calculated an eight-year median of the sediment yield for each individual watershed. Second, we spatially
aggregated the watersheds based on their dominant erosion characteristics (field- and structure-dominated)
while maintaining an annual resolution. Third, we aggregated the median values over the eight-year monitoring
period for these spatially aggregated groups.
To further analyse relative errors, the percent bias ($PBIAS$) was calculated by:
$$PBIAS = \left(\frac{Sim_i - Obs_i}{Obs_i}\right) * 100, \tag{8}$$
Where $Sim_i$ is the simulated values for behavioural realisation $i$ (t ha$^{-1}$ yr$^{-1}$), and $Obs_i$ is the observed sediment
value for realisation $i$ (t ha$^{-1}$ yr$^{-1}$).
**3. Results**
**3.1 Model Performance Across Scales**
The annual model results for field-dominated watersheds (W01-W04) were within the same order of magnitude
of the measured sediment yields. However, the model was not considered behavioural for predicting annual
sediment yields according to our pre-established acceptability criterion. The simulated annual sediment yields
were predominantly overestimated (22 out of 32 cases; Fig. 3a-d), occasionally underestimated (3 out of 32 cases,
i.e. in the year 2000 in W01; 1994 in W02; 1994 in W04; Fig. 3a, b and d), with only a small portion of simulations
meeting our acceptability criterion (7 out of 32 cases; Fig. 3a-d). The tendency to overestimate sediment yield is
more pronounced in watersheds W05 and W06. Only in 1994 the model underestimated measured sediment



yields in watershed W05 (Fig. 4b). In W06, measured sediment yields were the lowest among all watersheds
(maximum of 0.02 t ha⁻¹ yr⁻¹ in 2000), with zero sediment yield measurements in 1995, 1997, and 2001, yet the
model consistently overestimated sediment yield across all years in this watershed.

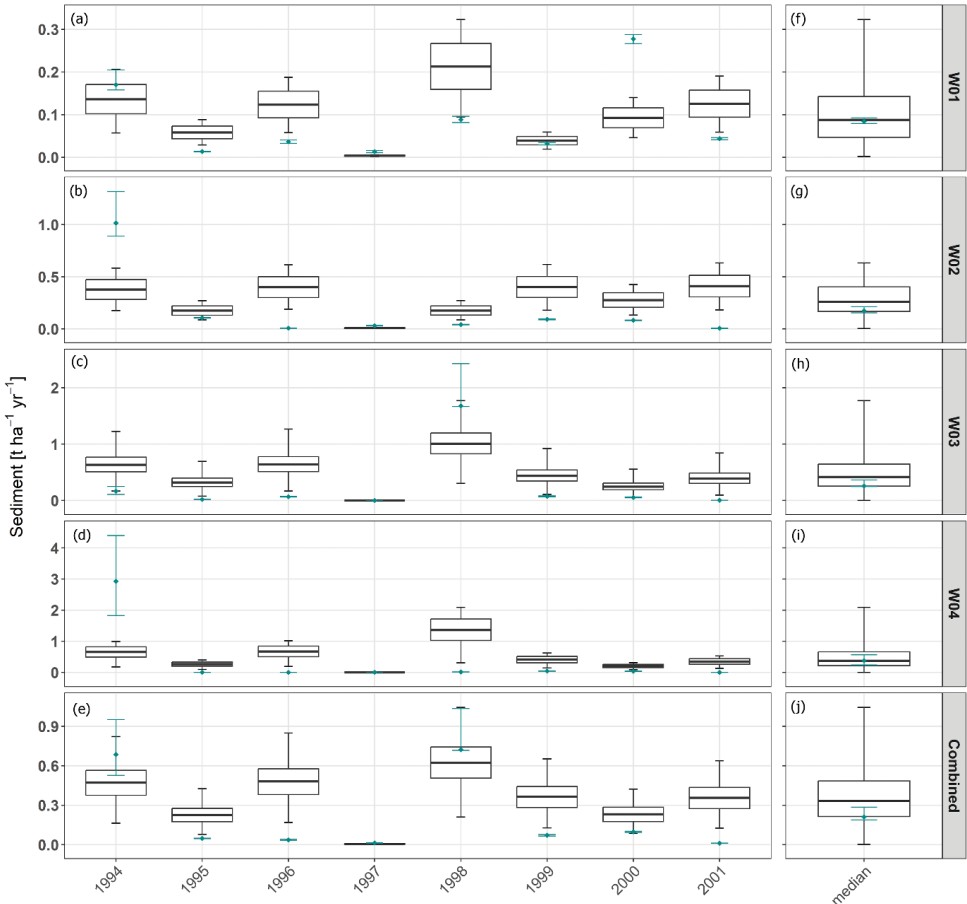


**Figure 3: Annual and eight-years median sediment yields in field-dominated watersheds. (a-d) Box plots display the median, 1. and 3. quartile and the full range of simulated sediment yields from 25,000 model realisations with different parameter sets (black whiskers), while median sediment yield measurements are shown as blue dots with computed error ranges (cyan whiskers). (f-i) The watershed-specific 8-year median measured yield with error ranges and simulated yields. (e) Annual spatially combined watershed sediment yields. (j) 8-year median of spatially aggregated watersheds.**

When evaluated using eight-year median values, model performance showed better agreement with
observations. The eight-year median modelled sediment yield across field-dominated watersheds (W01-W04)
was 0.24 t ha⁻¹ yr⁻¹, closely aligning with the measured eight-year median of 0.21 t ha⁻¹ yr⁻¹. For structure-
dominated watersheds W05 and W06, we simulated an eight-year median of 0.15 t ha⁻¹ yr⁻¹ (Fig. 4c, d), against a
measured median of 0.13 t ha⁻¹ yr⁻¹. The 1994 sediment yield peak in W05 strongly influenced the system's overall
performance, ultimately leading to an increased number of behavioural model realisations when evaluated across
the entire period (Fig. 4d).



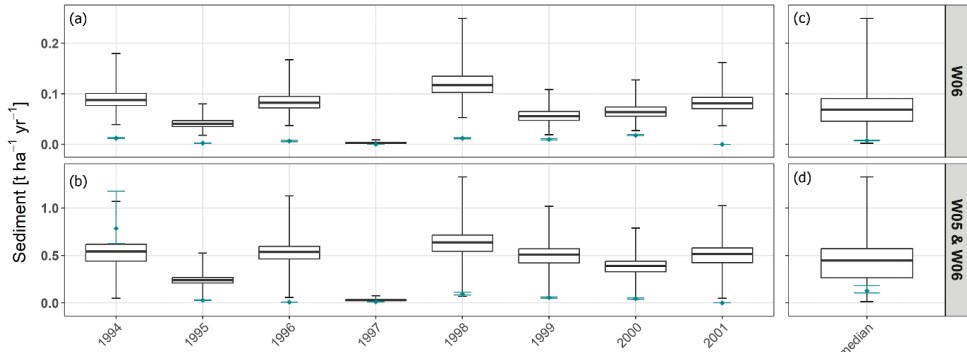


**Figure 4: Annual and eight-years median sediment yields in structure-dominated watersheds. (a-b) Box plots display the median, 1. and 3. quartile and the full range of simulated sediment yields from 25,000 model realisations with different parameter sets (black whiskers), while median sediment yield measurements are shown as blue dots with computed error ranges (cyan whiskers). (c-d) The watershed-specific 8-year median measured yield with error ranges and simulated yields.**

The eight-year temporal aggregation revealed varying proportions of behavioural model realisations across
individual watersheds. W04 had the highest amount with 57 % of all realisations, while other watersheds
exhibited lower proportions (W01: 13 %, W02: 21 %, W03: 23 %). W05 exhibited minimal behavioural realisations
of 1 %. Although the range of our simulated values overlapped with the error margins for the measured sediment
yields in W06 (Fig. 4c), none of the actual model realisations matched the observational data including
measurement errors. Furthermore, no common behavioural realisations were found across all watersheds,
indicating that each watershed had a different behavioural parameter space.
The analysis of the spatially aggregated watersheds (field-dominated vs. structure-dominated), while maintaining
annual temporal resolution, revealed behavioural model realisations in some years but not consistently
throughout the entire eight-year period for each watershed group (Fig. 3e, 4b). When combining both spatial and
temporal aggregation, behavioural realisations were generated for each watershed group (Fig. 3j, 4d). Across the
entire set of 25,000 realisations, the median $MAE$ values were 0.12 t ha$^{-1}$ yr$^{-1}$ for field-dominated watersheds and
0.16 t ha$^{-1}$ yr$^{-1}$ for structure-dominated watersheds, with maximum $MAE$ values of 0.34 t ha$^{-1}$ yr$^{-1}$ and 0.39 t ha$^{-1}$
yr$^{-1}$, respectively. Table 4 presents the model performance metrics specifically for the subset of behavioural model
realisations within the watershed groups.








**Table 2: Comparison of model performance metrics between micro-scale watershed groups based on eight-year median of**
**behavioural realisations, including median sediment yield (SY) as well as error statistics (*MAE*, *PBIAS*) with maximum**
**(Max.), median (Med.) and minimum (Min.) values.**

| Unit of measure | | Field-dominated | Structure-dominated |
|---|---|---|---|
| Behavioural realisations [%] | | 30.04 | 1.33 |
| Measured SY [t ha$^{-1}$ yr$^{-1}$] | Med. | 0.21 | 0.13 |
| Simulated SY [t ha$^{-1}$ yr$^{-1}$] | Med. | 0.24 | 0.15 |
| *MAE* [t ha$^{-1}$ yr$^{-1}$] | Min. | $4.21 \times 10^{-6}$ | $5.76 \times 10^{-5}$ |
| | Med. | 0.03 | 0.03 |
| | Max. | 0.07 | 0.05 |
| *PBIAS* [%] | Min. | -10.79 | -17.70 |
| | Med. | 15.35 | 20.15 |
| | Max. | 35.38 | 42.64 |

**3.2 Behavioural parameter space**
We analysed the behavioural parameter space for the spatially and temporally aggregated watershed groups, as
only this aggregation level yielded behavioural realisations for both field-dominated and structure-dominated
watershed groups.
For field-dominated watersheds, the analysis focused on the error surface and in-field parameter $e_{sur}$ and $k_{TC/A}$.
While behavioural realisations were identified across the entire ranges of all parameters, higher likelihood values
concentrated in specific regions. Specifically, $e_{sur}$ values closer to -0.5 exhibited higher likelihood values than lower
$e_{sur}$ values (Fig. 5b). In contrast, $k_{TC/A}$ showed no discernible pattern across the response surface (Fig. 5c). The
relationship between these parameters revealed a clear compensation mechanism, where lower $k_{TC/A}$ values
required higher $s_{cor}$ values to produce behavioural realisations (Fig. 5a).

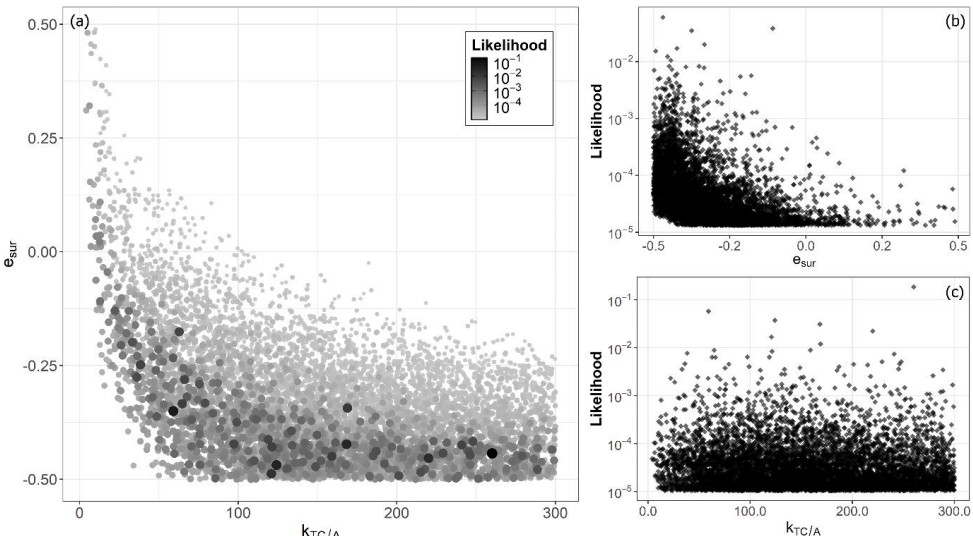

**Figure 5: Parameter likelihoods across field-dominated micro-scale watersheds, showing only behavioural model realisations. (a) The relationship between $e_{sur}$ and $k_{TC/A}$ parameters. Circle size and shade intensity indicate the likelihood of each parameter combination, with larger and darker circles representing higher likelihood values. (b) The relationship between likelihood and $e_{sur}$. (c) The relationship between likelihood and $k_{TC/A}$.**

In structure-dominated watersheds, the analysis focused on parameters controlling sediment transportation and deposition in grassland ($k_{TC/G}$, $b_{dep}$, and $p_{con}$). The $k_{TC/G}$ parameter exhibited a distinct likelihood peak between approximately 9 and 11 m, with behavioural values ranging from approximately 7.5 m to 15 m (Fig. 6a), which is notably narrower than the sampled range of up to 150 (Tab. 1). In contrast, $b_{dep}$ and $p_{con}$ showed no sensitivity, displaying relatively uniform likelihood distributions across their entire ranges (Fig. 6b, c). When plotting $b_{dep}$ against $p_{con}$, homogeneous likelihood distributions emerged with no apparent dependencies (Fig. 6f). Examinations of $b_{dep}$ and $p_{con}$ against $k_{TC/G}$ revealed a horizontal band of high likelihood values at specific $k_{TC/G}$ values, without any directional trends (Fig. 6d, e).



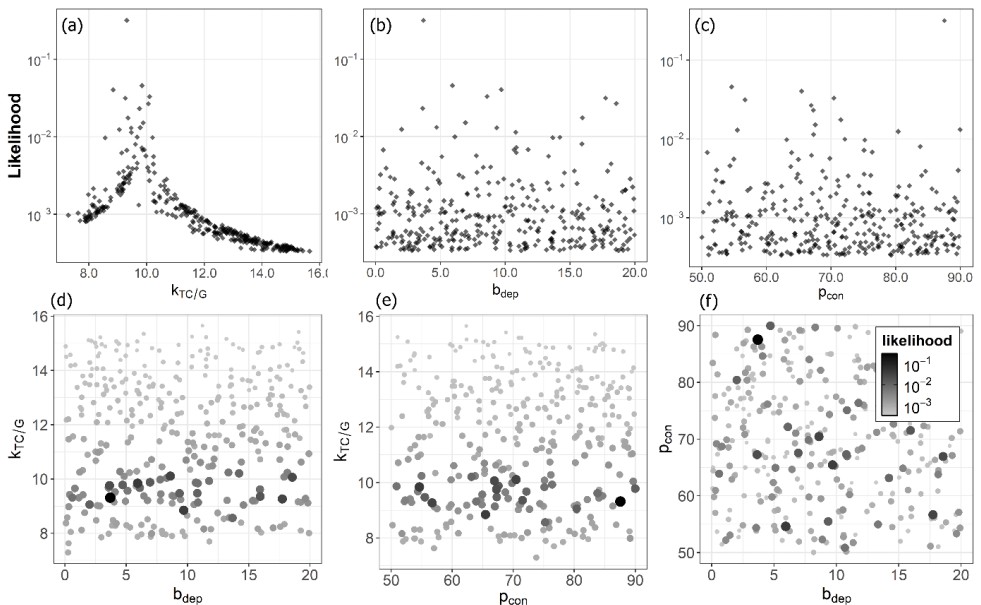

400

**Figure 6: Parameter likelihoods across structure-dominated micro-scale watersheds, showing only behavioural model realisations. (a) The relationship between likelihood and $k_{TC/G}$. (b) The relationship between likelihood and $b_{dep}$. (c) The relationship between likelihood and $p_{con}$. (d) The relationship between $b_{dep}$ and $k_{TC/G}$. Circle size and colour intensity indicate the likelihood of each parameter combination, with larger and darker circles representing higher likelihood values. (e) The relationship between $p_{con}$ and $k_{TC/G}$. (f) The relationship between $b_{dep}$ and $k_{TC/G}$.**

**3.3 Spatial analysis**

In field-dominated watersheds, substantial deposition was primarily confined to retention ponds, while other areas outside arable lands showed relatively minimal deposition, as shown in the 50th percentile (median) of behavioural model realisations (Fig. 7a). In W04, negligible to no deposition was observed. Conversely, structure-dominated watersheds exhibited considerably more intense erosion-deposition dynamics. The grassed waterway



showed a clear deposition pattern, with W06 exhibiting the most pronounced deposition patterns leading toward
the retention pond at the outlet of W06.

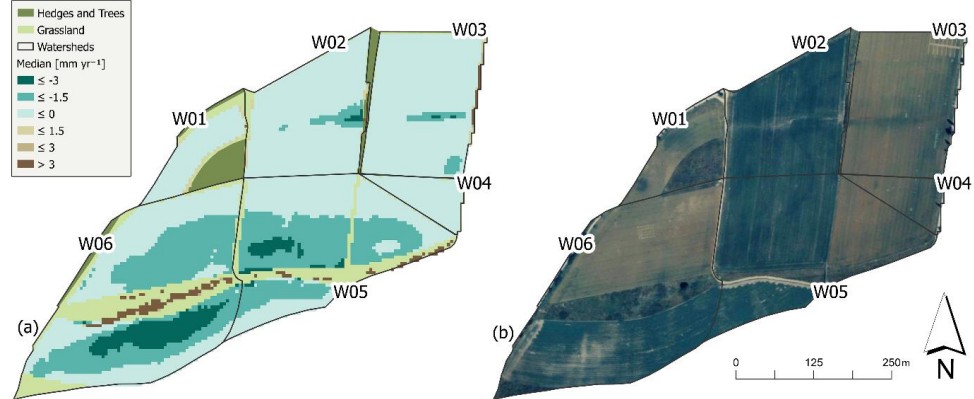


**Figure 7: (a) The median of simulated potential erosion and deposition of behavioural model realisations over the eight-**
**year period. (b) An aerial photograph of the study area shows the land use patterns and field boundaries on the Scheyern**
**experimental farm in 2002.**


**4. Discussion**
**4.1 GLUE Framework and Uncertainties**
We tested WaTEM/SEDEM using a limits of acceptability approach within the GLUE framework. For this, we
implemented a two-phase approach, first conditioning and evaluating field-dominated watersheds, and then
using the behavioural parameter space of these watersheds to condition and evaluate structure-dominated
systems. Alatorre et al. (2010) demonstrated that soil erosion models often exhibit parameter compensation
effects, where different parameter combinations produce similar outputs at the catchment scale - a manifestation
of the equifinality concept (Beven, 2006). Our sequential approach helped to minimise these effects by first
constraining the simulated erosion ($k_{TC/A}$ and $e_{sur}$) in field-dominated watersheds and then conditioning the
transport parameters ($k_{TC/G}$, $p_{con}$ and $b_{dep}$) in more complex systems.
The limits of acceptability approach incorporated multiple sources of measurement uncertainty. Nearing (2000)
demonstrated through replicated plot studies that natural variability in erosion measurements is particularly
pronounced for low-magnitude erosion events, such as the ones observed in this study. While Nearing (2000)
proposed a quantitative method for estimating the expected variability of erosion measurements, his approach
is specifically developed for plot-scale studies and cannot be extrapolated to watersheds or more complex
landscape systems. Given the (to the best of our knowledge) current absence of methodologies for determining
error boundaries for low sediment yield measurements at larger scales, we necessarily relied on relative error
estimates. An implementation of proper measurement variability-derived error ranges would likely result in a
substantially higher number of behavioural model realisations, particularly for low sediment yield measurements
where the variability is the highest (Nearing, 2000). This reveals the need for developing robust approaches for





defining limits-of-acceptability criteria for sediment yield estimates that account for the full range of
uncertainties, e.g. instrument precision, sampling errors, data processing, or site-specific variations.
Erosion models typically exhibit systematic biases, overpredicting low sediment yields while underpredicting high
sediment yields (Nearing, 1998; Risse et al., 1993; Kinnell, 2007). This is particularly relevant for our study area,
where the median measured sediment yield of 0.16 t ha$^{-1}$ yr$^{-1}$ was substantially lower than erosion rates which
can exceed 10 t ha$^{-1}$ yr$^{-1}$ in the Bavarian Tertiary hill region (Auerswald et al., 2009). To investigate the modelling
under/over prediction issue, we used an error surface ($e_{sur}$) multiplied with the erosion calculated by the USLE
(Eq. 5). The $e_{sur}$ parameter served three purposes: (i) adjusting potential erosion to investigate the USLE's inherent
biases, (ii) analysing the biases by looking at the behaviour of $e_{sur}$, and (iii) representing the uncertainty stemming
from measurement errors for the USLE factors and the lack of parameterisation for the P factor. The analysis of
behavioural model realisations revealed a concentration of likelihood values near small $e_{sur}$ values, reducing
sediment by up to 50 % (Fig. 5b). This indicates that in our study WaTEM/SEDEM overestimates soil erosion in
landscapes with implemented conservation measures. This is also evident looking at figure 3a-d and 4a-b, which
illustrate a general tendency for overestimation of modelled sediment yields in all watersheds.
**4.2 Model Performance and Limitations**
WaTEM/SEDEM correctly simulated the magnitude of the very low sediment yields in micro-scale watersheds
under optimized soil conservation, with annual values closely aligning with measured data (Fig. 3a-d, 4a-b).
Despite this achievement, the model did not consistently meet our strict limits of acceptability for annual
realisations and therefore was rejected for making precise annual simulations. However, the model's performance
improved notably when applied to longer-term medians and larger spatial units, where more behavioural model
realisations were identified.
**Field-dominated watersheds**
The model simulated the very low sediment yields resulting from well-established in-field soil conservation
practices in field-dominated watersheds, comparable to the measured data. In general, observed sediment yields
were overestimated, which can be attributed primarily to difficulties in accurately representing the specific C
factors of this conservation system, particularly unique practices such as mustard sown onto autumn-built dams
where potatoes were later directly planted (Fiener and Auerswald, 2003). Such unconventional approaches are
not adequately captured in the *SLR* values for no-till systems as evaluated in the German adaptation of the USLE
(ABAG; Schwertmann et al., 1987; Naw, 2022), even with the use of very low soil loss ratios in the
parameterisation of the C factor, which represent the continuous soil cover through the crop rotation in the
experimental farm (Fig. 2).
Conversely, the model underestimated sediment yields in some years because even optimally managed
conservation systems experience short time windows with weak protection. During these short windows with
reduced soil protection (Fig. 2), substantial erosion events may occur, like in systems not under soil conservation.
In general, erosion processes are typically dominated by extreme events (Gonzalez-Hidalgo et al., 2012; Steegen
et al., 2000), as exemplified in our study by an April 1994 rainfall event of 114 mm within 66 hours coinciding with



low soil coverage in W02 (Fig. 2), accounting for approximately 58 % of that year's total sediment yield (see year
1994 in Fig. 3b). The model's annual time step fails to capture these critical temporal coincidences, a limitation
that becomes more pronounced when such events are infrequent. This temporal limitation aligns with findings
by Risse et al. (1993), who demonstrated that USLE's model efficiency diminishes at the annual scale. When
averaging over the eight-year study period, these extreme events are smoothed out, which explains the model's
improved performance at longer timescales (Tab. 2). This observation supports the basic assertion that the USLE
was designed to compute long-term soil losses (Wischmeier and Smith 1978).
For the temporally aggregated eight-year medians, there was no single parameter set that produced behavioural
model realisations across all field-dominated watersheds simultaneously when applying our limits of acceptability
criterion. This indicates a limitation in parameter transferability within our study context. While Van Rompaey et
al. (2001) recognized technical limitations of WaTEM/ SEDEM in model transferability related to grid size and
routing methods, our findings suggest additional challenges in accurately representing processes within micro-
scale conservation landscapes. The need for watershed-specific calibration, even within relatively homogeneous
landscapes with similar crop and soil properties, indicates that parameter calibration compensates for inherent
model or data limitations. At such fine scales, WaTEM/SEDEM may struggle to accurately represent the complex
interactions between soil conservation measures and erosion processes.
Similar calibration challenges seem to exist more broadly in WaTEM/SEDEM applications across different
landscape types and research questions, as evidenced by a wide range of calibrated $k_{TC}$ values reported across
different studies (Tab. 3), with $k_{TC/A}$ values varying from 10 to 174.4 m. As Beven (2006) argues, such calibration
approaches may achieve mathematical fitting while concealing fundamental model inadequacies.
**Structure-dominated watersheds**
Unlike field-dominated watersheds, structure-dominated systems demonstrated different response patterns to
extreme erosion events. In these watersheds, sediment generated during individual large erosion events (as
observed in the field-dominated watersheds), is predominantly captured by grassed waterways and retention
ponds (Fiener and Auerswald, 2003, 2005), thus reducing the variability of event sediment yields. This buffering
effect explains why the model consistently overestimates sediment yield across all years for structure-dominated
watersheds (Fig. 4a-b), in contrast to the occasional underestimation observed in field-dominated systems (Fig.
3a-d).
Only one exception to this pattern was observed: the model underestimated sediment yield in W05 during 1994,
when the lower part of the grassed waterway required reseeding after losing its initial grass cover along the
thalweg during a spring erosion event (Fiener and Auerswald, 2003). This exceptional case quantitatively
demonstrates the role of functional grassed waterways, as the measured sediment yield in 1994 for W06 was
substantially higher (0.78 t ha$^{-1}$ yr$^{-1}$) than in subsequent years when the grassed waterway was fully established
(averaging only 0.03 t ha$^{-1}$ yr$^{-1}$ from 1995-2001), representing an approximately 96 % reduction in sediment yield
(Fig. 4b).



The model's systematic overestimation of sediment yields in structure-dominated watersheds reveals limitations
in representing the sediment trapping mechanisms of grassed waterways. The primary limitation is the model's
inability to capture re-infiltration processes within the grassed waterway, which is not accounted for in
WaTEM/SEDEM's transport capacity formulation. Fiener and Auerswald (2005) demonstrated that grassed
waterway effectiveness depends strongly on morphological characteristics, particularly the cross-sectional shape,
with flat-bottomed waterways showing substantially higher runoff reduction. The infiltration increases with
length and flatter cross-section of grassed waterways, which provide larger runoff widths and consequently
greater infiltration areas. A previous study showed that in the upper part of the grassed waterway (W06), where
WaTEM/SEDEM more substantially underestimates the sediment trapping, the long-term runoff and sediment
yield reduction was about 90% and 97%, while it was about 10% and 77% in the lower part of the grassed
waterway (W05) with a ditch-like cross-section (Fiener & Auerswald, 2003).
**4.3 Distribution of behavioural model parameter values**
The *TC* within agricultural fields is primarily controlled by a high transport coefficient $k_{TC/A}$ (Van Rompaey et al.,
2001). Lower $k_{TC/A}$ values reduce TC, promoting in-field deposition and consequently decreasing sediment yield
at the watershed outlet. Our analysis revealed behavioural model realisations across the full *a priori* selected
range of $k_{TC/A}$ values, with no clear pattern for field-dominated watersheds, demonstrating no sensitivity even at
very low $k_{TC/A}$ values near 1 or very high $e_{sur}$ values of 0.5 (Fig. 5a). This lack of sensitivity may be attributed to the
implementation of retention ponds in W01 and W02 and by the very low simulated erosion values, as *TC*
remained sufficiently high to transport the generally low sediment fluxes even with very low $k_{TC/A}$ values.
The low transport capacity coefficient for rougher surfaces, in case of this study for grassland $k_{TC/G}$, usually triggers
deposition in these areas (Van Rompaey et al., 2001). Our analysis identified behavioural values for $k_{TC/G}$ between
approximately 7.5 m to 15 m with a notable likelihood spike between approximately 9 m and 11 m, relatively low
values compared to other studies (Tab. 3). While Onnen et al. (2019) reported similarly low values for Danish
landscapes, they explicitly attributed this to sandy soils in Denmark. However, our study area features
predominantly silt loam and loamy soils, which are much more comparable to the Belgian soils (Tab. 3) where
low $k_{TC}$ values for rough surfaces were implemented (Peeters et al., 2008; Verstraeten et al., 2002; Van Rompaey
et al., 2001).
**Table 3: Comparison of $k_{TC}$ parameter values of behavioural model realisations used in different studies with**
**WaTEM/ SEDEM.**

| High $k_{TC}$ values mostly used for arable land [m] | Low $k_{TC}$ values mostly used for non-arable land [m] | Country | Source |
|---|---|---|---|
| 150 | not used | Germany | Wilken et al. (2020) |
| 10 to 24 | 1 to 12 | Denmark | Onnen et al. (2019) |
| 100 & 150 | 25 | Belgium | Peeters et al. (2008) |
| 75 | 42 | Belgium | Verstraeten et al. (2002) |
| 75 | 42 | Belgium | Van Rompaey et al. (2001) |
| 174.4 | not used | Belgium | Van Oost et al. (2000) |




These low $k_{TC/G}$ values can have some possible interpretations: (i) most likely, the model is compensating for its
inability to represent re-infiltration processes in grassed waterways, and/or (ii) the model may partly compensate
for an overestimation of erosion rates in the draining fields. However, although the model outputs are fully
spatially distributed (Fig. 7a), it is not possible to compare the simulated patterns with spatially distributed
observational data (e.g. aerial images, field surveys), because, except for some rare larger events, the effective
soil conservation established prevents visible erosion features like rills.
In our study, WaTEM/SEDEM showed no sensitivity to parameters $b_{dep}$ and $p_{con}$ that represent the influence of
linear landscape features. These parameters displayed homogenous likelihood distributions across the sampled
parameter space (Fig. 6b-c). This lack of sensitivity could stem from several factors: (i) sampling an overly narrow
parameter space, (ii) limited influence of field borders in the studied watersheds due to the layout of the fields
and watersheds with a small number of border situations, and/or (iii) a dominance of $k_{TC/G}$ implemented over a
long grass structure, which may nullify the influence of $b_{dep}$ and $p_{con}$ in the model outputs, especially in watershed
W05 and W06.
An additional limitation of the current parameterisation approach is its static nature. The effectiveness of grassed
waterways and retention structures varies throughout the year due to seasonal vegetation changes (Fiener and
Auerswald, 2003). Additionally, there is an important interaction between sediment influx and trapping
efficiency—as influx increases, the relative trapping efficiency typically decreases (Dermisis et al., 2010; Fiener
and Auerswald, 2018). Dermisis et al. (2010) demonstrated this inverse relationship, showing that grassed
waterway trapping efficiency decreases as peak runoff discharge increases, with notable breakpoints in efficiency
between different flow rates. The current static connectivity and transport capacity parameters ($p_{con}$, $b_{dep}$ and
$k_{TC/G}$) cannot adequately capture these temporal variations and flux-dependent relationships, suggesting the need
for a more dynamic parameterisation approach that accounts for both seasonal changes and influx response.
**5. Conclusion**
We evaluated WaTEM/SEDEM's capability to simulate sediment yields in micro-scale watersheds under optimised
soil conservation practices using a limits-of-acceptability approach within the GLUE framework. Our investigation
examined model performance across different levels of spatiotemporal data aggregation and analysed the
sensitivity of the model's response surface to the variability in the behavioural parameter space. Moreover, we
used a two-step conditioning process, in which model parameters linked to in-field erosion processes were
conditioned in field-dominated watersheds and later applied in structure-dominated watersheds, for which a
separate set of connectivity parameters was also conditioned.
The model was unable to produce behavioural realisations at annual timesteps based on our strict limits of
acceptability criterion despite the small absolute prediction errors (eight-year $MAE$ = 0.12 t ha$^{-1}$ yr$^{-1}$ for field-
dominated and eight-year $MAE$ = 0.16 t ha$^{-1}$ yr$^{-1}$ for structure-dominated watersheds). For the field-dominated
watersheds, the model particularly struggled with the simulation of annual sediment yields when individual
extreme events dominated the annual sediment production.



Aggregating model outputs in time and space worked best for field-dominated systems, which compensated for
the underestimation of soil conservation in controlling soil erosion and the model's inability to capture extreme
events within an annual time step. This finding confirms that WaTEM/SEDEM is better suited for long-term
conservation planning than for making precise annual sediment yield predictions in areas with soil conservation
practices.
The GLUE framework revealed specific patterns in the sampled parameter space, particularly the compensation
mechanism between $k_{TC/A}$ and $e_{sur}$ values for field-dominated watersheds, and the narrow behavioural parameter
range of $k_{TC/G}$ values (7.5-15 m) for structure-dominated watersheds. The likelihood distributions of $k_{TC/A}$ and
especially $e_{sur}$ enabled the pre-conditioning of structure-dominated watersheds, reducing parameter
compensation effects that typically mask model structural deficiencies.
Ultimately, our study demonstrates that WaTEM/SEDEM can simulate the very low sediment yields observed
from soil conservation agricultural systems, provided that high spatiotemporal resolution input data and locally
adapted USLE factors (e.g., the ABAG for Southern Germany) are available. However, capturing the effects of
linear landscape features like grassed waterways where concentrated runoff occurs remains challenging for
WaTEM/SEDEM, primarily due to the model's inability to represent re-infiltrating processes that are critical for
sediment trapping in such structures. Additionally, our model evaluation approach revealed that model
performance strongly depends on the spatiotemporal scale of analysis. While the model produced behavioural
realisations for the aggregated eight-year monitoring period, it did not reliably simulate annual sediment yields.
For long-term, large-scale soil conservation planning in which the effects of single erosive events on individual
fields are less relevant for representing the system behaviour, WaTEM/SEDEM is fit for purpose.
**Code availability**
The code is available on reasonable request.
**Data availability**
The    input    data    are    openly    available    and    can    be    downloaded    here:
https://adgeo.copernicus.org/articles/48/31/2019/adgeo-48-31-2019.html (Last visited 11.07.2025)
**Author contribution**
Kay Seufferheld: Conceptualisation, Data curation, Formal analysis, Investigation, Methodology,
Software, Visualisation, Writing - Original Draft; Pedro V. G. Batista: Conceptualisation, Methodology,
Validation, Writing (review and editing); Hadi Shokati: Writing (review and editing); Thomas Scholten:
Supervision, Writing - Review & Editing; Peter Fiener: Conceptualisation, Data curation, Project
administration, Supervision, Validation, Writing (review and editing).



**Competing interests**
Pedro V. G. Batista and Peter Fiener serve as Topic Editor and Executive Editor of SOIL, respectively.
**Acknowledgments**
We acknowledge the valuable dataset from the Forschungsverbund Agrarökosysteme München (FAM). The
scientific activities of the FAM research network were financially supported by the German Federal Ministry of
Education and Research (BMBF 0339370). We thank all researchers and technical staff involved in collecting and
maintaining these long-term datasets, which made this study possible.
During the preparation of this work, the authors used Anthropic Claude 3.7 Sonnet to improve the readability
and language of the manuscript. After using this tool, the authors reviewed and edited the content as needed
and take full responsibility for the content of the published article.
**Financial support**
This research was funded by the German Research Foundation (DFG) through the DYLAMUST project (Project
number: 509809226).

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
