# Peer review of "A GLUE-based assessment of WaTEM/SEDEM for simulating soil"

_EGUsphere, 2025_

## Author Comment (AC1)

Thank you for the for the detailed review of our manuscript. We appreciate the constructive feedback, which we believe will improve our work. We will provide a point-by-point reply to all your comments in our final response letter, but we would like to take the opportunity address your main concerns while the interactive discussion is open.

Below the major comments from referee #1 are displayed in orange, and the author response in blue:

**Referee comment:**

In terms of the scientific advancement which are offered for soil erosion modelling, I find the conclusion to reiterate what is perhaps the most fundamental concept of the USLE - that temporal aggregation is required to generate acceptable predictions according to the central limit theorem - which isn't neccessarily surprising. Reiterating this point is nevertheless useful, however one could argue that "why" is more relevant than "if". While the study correctly demonstrates that the model fails at the annual timestep, it offers limited insight into the reasons for this failure. A key value of testing a model outside its intended use case is to diagnose its structural weaknesses; this potential is not fully realised here.

**Author's response:**

We agree that reiterating the fact that USLE-based models are designed for predicting long-term soil losses is an important and worthwhile point. We also agree that the "why" is more important than the "if", in this case. We apologise if the insights for model rejection at annual timesteps were insufficiently developed. This was discussed our preprint, where we explain that for the in field-dominated watersheds:

(i)     over estimation of sediment yields at annual timesteps was very likely associated to the uniqueness of the soil conservation practices employed at the Scheyern experimental farm (e.g. sowing mustard onto autumn-built dams where later potatoes were directly planted), which are difficult to represent within model parameterisation; (L460-467)

(ii)    under estimation of sediment yields is linked to the occurrence of high magnitude rainfall events during very rare time windows of low soil cover, which disproportionally affect sediment yields on annual time scales, albeit this effect is diluted in the long term. (L468-475)

In the structure dominated watersheds, we explain that lower prediction accuracy of sediment yields on annual timesteps is due to:

(i)     lack of processes representation in the model (i.e. re-infiltration along grassed waterways); (L510-515)

(iii)   the occurrence of a rainfall event after the reseeding of the grassed waterway that lost its initial grass cover along the thalweg during a spring erosion event (L501-507).

In our revision, we will organise the manuscript in a way that these points are clearer and will also include a discussion on the matter of spatial aggregation.

We would also like to highlight that the value of our work goes beyond reiterating temporal issues with USLE-based models, as we (i) demonstrate a case of model evaluation using a rejectionist framework considering uncertainty in models and the forcing data (which is lacking in soil erosion research), and (ii) we do so in soil conservation settings and in cascading, highly instrumentalised micro watersheds, in which soil erosion and sediment delivery models are poorly constrained. We

will emphasise the importance of our contribution in the revised manuscript, so that this is clearer to readers.

**Referee comment:**

Linking to this broader point is the main methodological critique I have of the manuscript. Despite using an implementation design which is intended to understand model drawbacks and accept only a subset of simulations, the authors do not consider uncertainty on the individual USLE parameters but instead opt for the use of an error surface on the gross erosion predictions. Given that a subset of these parameters are propagated into the transport capacity (i.e. L, S, R and K), the setup potentially ignores important parameter interactions which may impact both the uncertainty quantification and the insights into the model's shortcomings. Considering the simplistic design of the model, parameter lumping seems avoidable.

**Author's response:**

Thank you for this critical methodological point. You are correct that we opted to lump the uncertainty from the ABAG factors into a single error surface rather than sampling each factor in the Monte Carlo simulation. This was a methodological choice driven by two primary considerations: (i) our study's specific objectives and data quality, (ii) computational costs:

(i)     Our primary objective focused on testing WaTEM/SEDEM's transport component within micro watersheds under soil conservation, and not the ABAG, which was developed for Southern Germany and where it has been extensively evaluated. Moreover, our study is based on an exceptionally high-quality monitoring dataset from the Scheyern experimental farm, which provides site-specific, highly detailed field measurements of rainfall, soil, and crop and management data that are used as input for calculating the ABAG factors (which were experimentally calibrated precisely for our study region - Schwertmann et al. (1987)). Hence, we assume that parameterisation errors are negligible, but as the ABAG is essentially built upon regressions that will always carry residual errors, we pragmatically used an error surface ($e_{sur}$) to account for this.

(ii)    Representing the uncertainty in each ABAG factor separately is not as straightforward as sampling different values in a Monte Carlo simulation. This is because these factors are not parameters being calibrated or conditioned against observations, but are more like variables that are estimated from input data. Uncertainty estimation in this case requires complex approaches for representing uncertainty in the input data used for calculating input factors and propagating them through the model (as we have done in previous work - Batista et al. (2021); Batista et al. (2022)). We did not feel like this was justified because of the quality of our input data, as we explain in point (i) above.

**Referee comment:**

The choice of approach also induces a circularity into the argumentation of the study, where the lack of acceptable model realisations at the annual timestep is attributed to unconsidered uncertainty in the input factors (e.g. the (bio)physical impacts of conservation tillage on overland flow and erosion), which could have otherwise been considered in the modelling approach.

**Author's response:**

Thank you for this point, but we disagree that our argument is circular. We give specific arguments to explain model rejection on annual timesteps (see responses above). It is also worth highlighting that the temporal aggregation works somewhat similarly to a sensitivity analysis in our framework,

as we can identify that temporally statistic factors (i.e. K and LS) cannot be driving the increase in model accuracy when results are aggregated into a longer time period. We will improve these arguments in our revised manuscript.

**Referee comment:**

Secondly, the study implements a replication of the WS code but performs neither benchmarking against the standard model nor releases the source code openly. I am in favour of replications of models such as WS in popular programming languages such as Python (which although on average slower, permit easy data integration and parallelization as mentioned by the authors), but without showing at least a benchmarking use-case the current implementation lacks reproducibility and good modelling practice.

**Author's response:**

Thank you for raising these important points, we are happy to address them:

(i) We did benchmark our Python code – apologies that this information was not included in the preprint. We conducted parallel tests using identical input data and parameter sets on both the original Delphi-based WaTEM/SEDEM and our new Python version for each step of the translation process. We confirmed that our Python implementation produces identical sediment yield outputs to the original model (within very small error ranges due to the rounding of values), ensuring that our replication is robust. We will include this information with further detail in the revised manuscript.

(ii) Our Python code is under active development by several group members and therefore not yet in a stage where we would feel comfortable publishing in permanent public archive. We have used the "available upon reasonable request" status as an interim solution. This is not to restrict access, but to ensure that anyone who requests the code receives the correct version along with the necessary guidance, which we are happy to provide. In any case, we will upload a version that will be accessible to the referees with the revised manuscript. To ensure the reproducibility of our specific findings in this study, we will include the R scripts used for statistical analysis, the data preparation code, and the raw model outputs as supplementary material to the revised manuscript. This allows for a verification of our data processing and statistical conclusions.

**References**

Batista, P. V. G., Fiener, P., Scheper, S., and Alewell, C.: A conceptual-model-based sediment connectivity assessment for patchy agricultural catchments, HYDROL EARTH SYST SC, 26, 3753-3770, 10.5194/hess-26-3753-2022, 2022.

Batista, P. V. G., Laceby, J. P., Davies, J., Carvalho, T. S., Tassinari, D., Silva, M. L. N., Curi, N., and Quinton, J. N.: A framework for testing large-scale distributed soil erosion and sediment delivery models: Dealing with uncertainty in models and the observational data, Environmental Modelling & Software, 137, 104961, 10.1016/j.envsoft.2021.104961, 2021.

Schwertmann, U., Vogl, W., and Kainz, M.: Bodenerosion durch Wasser: Vorhersage des Abtrags und Bewertung von Gegenmassnahmen, Stuttgart: Ulmer, 64 pp.1987.

---

## Author Comment (AC2)

Dear Joris,

Thank you for taking the time to review our manuscript and for the valuable comments and suggestions.

We will provide a point-by-point reply to all your comments in our final response letter, but we would like to take the opportunity address your main concerns while the interactive discussion is open.

Below your comments are displayed in orange, and the author response in blue:

**Referee comment:**

The objectives should be better defined. The first objective focusses on testing of the model's capabilities. This does not seem to be too ambitious. No matter what is the outcome, this objective will always be achieved. So please refine this objective to make it more ambitious. The second objective seems related to the GLUE approach, I suggest to explicitly include the GLUE approach in this objective. The last objective is similarly not too ambitious (either testing or analysing something will always be achieved).

**Author's response:**

This is a very good point – thanks for bringing it up.

We will refine the phrasing of our objectives to highlight the ambition and importance of our work. That is, our study employs a rejectionist limits-of-acceptability approach within the GLUE framework to test a widely used soil erosion and sediment delivery model in settings where it is routinely applied but not well evaluated against field data (particularly not high-quality, long-term monitoring data). In this context, "testing capability" is not a passive task where success is guaranteed, but a rigorous attempt at model falsification (Beven, 2006), which is critical to improve models and understanding, but rarely done in environmental modelling in general and erosion modelling in particular.

**Referee comment:**

The concept of aggregating the data using different spatiotemporal resolutions has not been mentioned in the Introduction. I was expecting that this would go in some direction of using different spatial and temporal resolutions (different cell sizes and time steps, for instance). However, this is totally not the case. The authors instead use the long-term median model outcome, instead of the annual outcomes (the way USLE-type of models should actually be used). And the spatial aggregation is related to the two different conservation types considered. I'm not sure if this requires to be included in an objective.

**Author's response:**

Apologies if this topic was not sufficiently developed in the introduction – we will make amends in the revised manuscript. There might be some misunderstandings we'd like to clarify:

(i)     Our analysis does compare two different temporal resolutions (perhaps we need to check the phrasing to make this more precise). We evaluate the model on annual timesteps (from 1994 to 2001) and then on the aggregated eight-year median. However, it is not correct that USLE-type models were designed to operate with an annual resolution. The original USLE was developed to predict long-term average annual soil loss, not to produce annual model outputs (Wischmeier and Smith (1978) - Page 2: "The USLE is an erosion model designed to predict the longtime average soil

losses in runoff from specific field areas in specified cropping and management systems."; or page 40: "The USLE is designed to predict longtime-average soil losses for specified conditions."). Our finding that the model performs better when aggregated over the eight-year study period is therefore fully consistent with the original design and purpose of the USLE (or ABAG in our case).

(ii)    We apologise if our spatial aggregation (single watersheds or combined as field- or structure-dominated) was not clear or gave you the impression we looked at different cell-sizes for the input data. The aggregation is not simply between two different conservation types. It is between: (i) Field-dominated watersheds (W01-W04), which are characterized by in-field soil conservation practices (e.g., no-till, crop rotation), retention ponds in W01 and W02 and small grass-strips. (ii) Structure-dominated watersheds (W05, W06), which include in-field soil conservation practices plus additional linear landscape structures (specifically, retention ponds and a big, grassed waterway) designed to trap sediment. This distinction is important for our two-phase GLUE approach, where we first condition in-field parameters on the field-dominated group (Phase 1) and then use those to condition the structural parameters on the structure-dominated group (Phase 2). This aggregation is also important to evaluate the transferability of model conditioning between watersheds. We will make these points clearer in the revised manuscript.

**References**

Beven, K.: A manifesto for the equifinality thesis, J HYDROL, 320, 18-36, 10.1016/j.jhydrol.2005.07.007, 2006.
Wischmeier, W. H. and Smith, D. D.: Predicting rainfall erosion losses: A guide to conservation planning, Agriculture Handbook,  537, Department of Agriculture, Science and Education Administration, United States, 65 pp.1978.